# Particulate 3D Hydrogels of Silk Fibroin-Pluronic to Deliver Curcumin for Infection-Free Wound Healing

**DOI:** 10.3390/biomimetics9080483

**Published:** 2024-08-10

**Authors:** Azin Khodaei, Narges Johari, Fatemeh Jahanmard, Leonardo Cecotto, Sadjad Khosravimelal, Hamid Reza Madaah Hosseini, Reza Bagheri, Ali Samadikuchaksaraei, Saber Amin Yavari

**Affiliations:** 1Department of Orthopedics, University Medical Center Utrecht, 3584 CX Utrecht, The Netherlands; 2Institute for Nanoscience and Nanotechnology, Sharif University of Technology, Tehran 14588-89694, Iran; 3Department of Materials Science and Engineering, Sharif University of Technology, Tehran 14588-89694, Iran; 4Materials Engineering Group, Golpayegan College of Engineering, Isfahan University of Technology, Golpayegan 87717-67498, Iran; 5Utrecht Institute for Pharmaceutical Sciences, Department of Pharmaceutics, Faculty of Science, Utrecht University, 3584 CG Utrecht, The Netherlands; 6Department of Medical Biotechnology, Faculty of Allied Medicine, Iran University of Medical Sciences, Tehran 14496-14535, Iran; 7Regenerative Medicine Utrecht, Utrecht University, 3584 CT Utrecht, The Netherlands

**Keywords:** Poloxamer 407, wound healing, hierarchical, MRSA, scaffolds

## Abstract

Skin is the largest protective tissue of the body and is at risk of damage. Hence, the design and development of wound dressing materials is key for tissue repair and regeneration. Although silk fibroin is a known biopolymer in tissue engineering, its degradation rate is not correlated with wound closure rate. To address this disadvantage, we mimicked the hierarchical structure of skin and also provided antibacterial properties; a hydrogel with globular structure consisting of silk fibroin, pluronic F127, and curcumin was developed. In this regard, the effect of pluronic and curcumin on the structural and mechanical properties of the hydrogel was studied. The results showed that curcumin affected the particle size, crystallinity, and ultimate elongation of the hydrogels. In vitro assays confirmed that the hydrogel containing curcumin is not cytotoxic while the diffused curcumin and pluronic provided a considerable bactericidal property against Methicillin-resistant *Staphylococcus aureus*. Interestingly, presence of pluronic caused more than a 99% reduction in planktonic and adherent bacteria in the curcumin-free hydrogel groups. Moreover, curcumin improved this number further and inhibited bacteria adhesion to prevent biofilm formation. Overall, the developed hydrogel showed the potential to be used for skin tissue regeneration.

## 1. Introduction

Skin is an exposed tissue that is vulnerable to various disorders such as partial/full-thickness burns, diabetic lesions, surgical injuries, trauma, and genetic defects [1]. According to the latest report of the World Health Organization (WHO), 265,000 deaths are reported annually due to burn wounds [2]. Although the cell proliferation rate in the skin is usually high [3], the intrinsic regeneration does not result in normal mimicked tissue, especially in cases of chronic disorders [4]. Skin grafts and scaffolds based on biomaterials have been developed over the last decades in attempts to replicate full-depth skin regeneration [5,6]. Many fabricated scaffolds for skin regeneration via methods like casting and electrospinning are 2D structures, However, full-depth regeneration needs a 3D scaffold [7,8].

Another important parameter that should be considered in designing skin scaffolds is water uptake. Water uptake prevents the skin from drying due to the high evaporation rate and also provides essential nutrients for the stem cells to proliferate and differentiate [9]. Hydrophilic polymers with micro/nano textures provide the highest water absorption ratio and are usually categorized as hydrogels. Hydrogels, therefore, are promising candidates for skin regeneration. However, high moisture at the site of the defect creates a suitable environment for bacterial growth and infection [4], which necessitates the use of hydrogels with antibacterial properties. The application of antibiotics, antimicrobial peptides, and inorganic antibacterial nanoparticles in the hydrogel has been widely studied but there are few publications on using surfactants to reduce planktonic and adherent bacteria growth [10]. Romic et al. showed that melatonin-loaded chitosan/pluronic F127 microparticles delivered higher antimicrobial and antibiofilm activity against *S. aureus* compared to melatonin-loaded chitosan [11]. Pluronic F127-conditioned polystyrene was also studied for C. albicans eradication by Wesenberg-Ward et al. [12]. They showed that F127 dramatically reduced adhesion and biofilm formation, which aligns with the conventional application of surfactants as wound cleansing agents. Besides, curcumin as a naturally derived antibacterial agent is known as an antibacterial, antioxidant, and anti-inflammatory agent and has long been used for wound dressing applications [13,14,15,16]. In skin tissue repair, it can also stimulate hair follicles for hair growth and inhibit scar formation during the healing procedure [17]. Panja et al. showed that encapsulation of curcumin in Bombyx mori SF prolonged the excited state of curcumin and consequently increased its therapeutic performance for biomedical applications [18].

Adding surfactants can also change the microstructure of biomaterials by inducing micellar and globular structures and homogenizing the distribution of hydrophobic drugs [19,20]. These structures can be beneficial for drug delivery systems as they provide open pores and facilitate the controlled release of drugs and reactive agents [21]. As skin is a multilayered tissue, a hierarchical porous scaffold mimics the native tissue very well. In fact, the suitable pore size for neovascularization, fibroblast ingrowth, and dermal repair was reported at 5, 5–15, and 20–125 μm, respectively [22], which shows the necessity of using hierarchical porous scaffold for skin tissue engineering. 

Silk fibroin (SF) is a biopolymer that has recently been used in tissue engineering as films, fibers, hydrogels, and 3D scaffolds [23,24,25,26]. SF hydrogels with high biocompatibility and controllable biodegradability can be prepared from the polymer solution after degumming [27]. As an advantage, the mechanical properties of SF-based scaffolds are adjustable with the degree of crystallinity [28]. Pure and blended SF hydrogels were used for deep regeneration of burned skins due to exudates absorption and their ability to cover irregular wounds [29]. In recent years, various blend hydrogels made from synthetic and natural hydrophilic polymers such as polyethylene, polyvinyl alcohol, agarose, gelatin, and hyaluronic acid have been developed to improve and manipulate the mechanical, structural, and biological properties of SF-based hydrogels [30,31,32,33,34,35]. 

In this study, a novel blend of silk fibroin/pluronic F127 hydrogel with a globular structure was developed to deliver curcumin. We aimed to correlate the degradation rate of SF and the proliferation rate of fibroblast cells by manipulating the microstructure of hydrogel. Furthermore, the mechanical properties and cellular and antibacterial behavior of the developed hydrogels are discussed. 

## 2. Materials and Methods

### 2.1. Silk Fibroin

Reconstituted SF solution was previously extracted by Johari et al. [36] according to the reported protocol by Kaplan et al. [30]. In the beginning, Bombyx mori cocoons (prepared from Iran Silkworm Research Center, Guilan, Iran) were boiled in 0.02 M aqueous sodium carbonate (Na_2_CO_3_, Merck, Darmstadt, Germany) solution for one hour as the degumming process. Then they were washed with cold and hot water several times to make sure of sericin extraction. In the following, the dried degummed silk was dissolved in a 9.3 M lithium bromide (LiBr, Merck, Germany) solution for 4 h at 55 °C, and the dialysis of the prepared solution was conducted at room temperature against Milli-Q water for 72 h. In this case, 12 kDa dialysis membrane was used. The final product was the aqueous SF solution (6% *w*/*v*). Pluronic F127 (Sigma-Aldrich, St. Louis, MO, USA), curcumin (Sigma-Aldrich, USA), and acetone (Sigma-Aldrich, USA) were used to fabricate the scaffolds. 

### 2.2. Synthesis of the Scaffolds

Three different blend solutions of SF/F127 with and without acetone/curcumin were prepared according to the compositions reported in Table 1. Pure silk fibroin was also used to prepare scaffolds as the control. The stock solution of curcumin was prepared by dissolving 10 mg/mL in acetone, and the stock solution of F127 was prepared 6 *w*/*v*%: g/mL. The curcumin solution was added to SF/F127 solution with a volume ratio of 7 *v*/*v*%: mL/mL and kept stirring for 30 min. The final solution was casted on NaCl crystals (1:1 *v*/*w*: mL/g) with an average diameter of 150–300 μm (sieved between 50 and 100 mesh). 

Every 4 g NaCl was placed in a 35 mm Petri dish and casted. After drying for three days at room temperature, the samples were washed three times with Milli-Q water to remove the salt crystals. The obtained gels were lyophilized at −50 °C and 0.05 mbar pressure overnight. 

To determine the potential cytotoxicity of pluronic, SFP samples with different ratios of SF/P: 2 ÷ 6 were also synthesized as previously mentioned. In this regard, the concentration of pluronic solution was adjusted to keep the concentration of SF constant in all the blended solutions. 

### 2.3. Characterization of Materials Properties

Morphology of the freeze-dried samples was studied using scanning electron microscopy (SEM, Phenom-pro desktop, ThermoFisher Scientific, Waltham, MA, USA). The samples were cut with a stainless-steel blade and were coated with an 8 nm layer of gold prior to microscopy. An accelerating voltage of 10 kV was employed in this study. ImageJ software was used to analyze the particle size in the micrographs. 

The chemical bonds in the samples were analyzed by Fourier Transform Infrared Spectroscopy (FTIR, MB-100, ABB Bomem, Quebec, Canada) within the wavenumber of 400–4000 cm^−1^. FTIR samples were prepared as KBr pellets. To evaluate the crystallinity of the samples, temperature-modulated differential scanning calorimetry (TMDSC-Q100/TA instruments, New Castle, DE, USA) was used. Nitrogen with a flow rate of 25 mL/min was purged as the inert media. Sapphire and indium were used to calibrate the apparatus. Amplitude A_T_ of 1.59 °C with a period of 60 s was the standard modulation condition. The temperature was ramped in the range of 180–250 °C, and the heating rate was adjusted at 3 °C/min. 

Dynamic mechanical analysis (DMA, TA instrument 2980, New Castle, DE, USA) was used to apply a parallel-plate compression test. A biopsy punch was used to prepare cylindrical samples of 6 mm diameter and height from freeze-dried samples. We tested the samples 24 h after they were immersed in PBS. The rate of force increase was 5 N/min up to the final force of 18,000 N. A linear fit was performed on the stress–strain data in the range of 0–10% strain to determine the compression modulus. All the quantitative data was collected by examining three replicates from each group sample.

### 2.4. Degradation, Swelling, and Drug Release

Degradation of the hydrogels was studied in the high-glucose Dulbecco’s Modified Eagle Medium (DMEM, Gibco, Thermo Fisher Scientific, Waltham, MA, USA). The samples were cut by a 6 mm biopsy punch. They were incubated in 200 medium at 37 °C and were collected after 3, 5, 7, and 14 days. After discarding the media, the samples were freeze-dried to weigh the remind material. The degraded suspension was also watched under an optical microscope (DMi1, Leica camera, Wetzlar, Germany). 

To evaluate the release profile of curcumin from SFP-A-Cur, the samples were incubated in phosphate-buffered saline (PBS) at 37 °C. To determine the curcumin concentration, the release medium was transferred to a 96-well-plate and was measured by a plate reader (Fluoroskan Ascent FL, Thermo Fisher Scientific, Waltham, MA, USA) while the absorption intensity of 400 nm was selected. All the group samples had three replicates to determine the standard deviation for the quantitative data. The calibration curve was prepared by diluting 10 mg/mL curcumin dissolved in acetone and diluted in PBS in the range of 0–75 μg/mL.

The swelling ratio of the samples after being immersed in PBS for five days was calculated based on dry and swelled weight of the samples. 

### 2.5. Cytotoxicity 

To evaluate the cytotoxicity of the hydrogels with different ratios of SF to pluronic (coded SF/P(2) to SF/P(6)), the MTT (3-[4,5-dimethylthiazol-2-yl]-2,5 diphenyl tetrazolium bromide) assay was performed 24, 48, and 72 h after the cell culture. Human dermal fibroblasts have been used for this purpose. In doing so, the samples were first punched using a 6 mm biopsy punch (weight of the samples is calculatable based on Table 1) and sterilized using UV exposure for 20 min. Afterwards, 10,000 cells were seeded on each sample and were incubated in full DMEM containing 10% Fetal Bovine Serum (FBS, Biowest, Nuaillé, France) and 1% Penicillin-Streptomycin (10,000 U/mL, ThermoFisher scientific, MA, USA) at 37 °C, 5% CO_2_ incubator. For the MTT assay, the media was replaced with 0.1 mL DMEM containing 0.02 mL MTT solution (5 µg/mL). The samples were incubated for 4 h until formazan crystals were formed. In this step, the reagent was removed, and 0.1 mL dimethyl sulphoxide (DMSO) was added to dissolve the formazan phase during 30 min of dark incubation at room temperature. Later, 100 µL of DMSO-based supernatant from each sample was transferred to a 96-well-plate to measure the absorbed optical density at 570 nm with a plate reader (Fluoroskan Ascent FL, ThermoFisher Scientific, Waltham, MA, USA). The control group in this experiment was the cell seeded on the polystyrene-well plate. The same procedure was also applied to evaluate the cytotoxicity of curcumin-loaded samples.

To assess the cell morphology and coverage of the samples, cells were fixed with 4% formalin in PBS at different time points. In the next step, cells were permeabilized with 0.2% Triton X-100 in PBS and were stained with tetramethylrhodamine B isothiocyanate (TRITC)-labeled phalloidin (2.5 µg/mL) and DAPI (2 µg/mL). Phalloidin stains actin (532–575 nm) and DAPI stains nuclei (405–480 nm). The stained samples were then imaged by a confocal laser scanning microscope (CLSM, SP8X, Leica Camera, Wetzlar, Germany).

### 2.6. Antibacterial Study

Methicillin-resistant *Staphylococcus aureus* strain (SH1000) containing a plasmid to constitutively express Green Fluorescent Protein (GFP) was used to determine the antibacterial properties of the hydrogels introduced in Table 1. Two direct and indirect methods of colony-forming-unit (CFU) testing and optical density (OD) measurements were used, respectively, to evaluate the number and growth of planktonic (non-adherent) bacteria. In the first step, the samples with the weight of 0.06 g were suspended in 2 mL Todd Hewitt broth (THB) and were incubated at 37 °C overnight. The supernatant was then collected and plated in a 96-well-plate. *S.aureus* was grown overnight in THB with 10 ng/mL chloramphenicol. The bacterial suspension was diluted in THB to reach a final inoculum concentration of 5 × 10^5^ CFU/mL and added to supernatants by a ratio of 1:1. The plate was incubated at 37 °C and bacterial growth was monitored by measuring OD (600 nm) continuously for 15 h on Clariostar plate reader (BMG Labtech, Ortenberg, Germany). Although the initial OD of different samples was different due to the degradation rate variation, the curves were adjusted on the same initial OD since the cultured concentration was the same for all the samples.

For the CFU test, the prepared bacterial suspension (1 mL) was seeded on the samples (0.03 g) and then incubated at 37 °C overnight. To determine the number of planktonic bacteria, the supernatant was serially diluted in PBS and plated on Todd Hewitt Agar (THA) plates. Samples were incubated at 37 °C overnight and colonies were counted the day after. To assess bacteria adhesion on the surface of the samples, the hydrogels were rinsed with PBS three times and then were sonicated in 2 mL PBS tubes for 60 s. Similar to the planktonic bacteria, the supernatant was diluted and plated for CFU count. The collected data in this test were normalized to the control group. The control group for planktonic bacteria is the free-floating bacteria in THB while the control group for adherent bacteria is the number of bacteria attached to the surface of SF. 

### 2.7. Statistical Analysis

All the statistical computation was performed using GraphPad Prism9 software (GraphPad Software, La Jolla, CA, USA) and the statistical significance was analyzed using one-way ANOVA with the post hoc Tukey test. The signs of *, **, ***, and **** in the graphs represent *p* < 0.05, 0.01, 0.001, and 0.0001, respectively. All the data are shown with standard deviation as the error bar. 

## 3. Results and Discussion

### 3.1. Hydrogel Characterization

SEM micrographs showed irregular pores around the diameter of NaCl crystals (150–300 µm) in all the group samples of SF, SFP, SFP-A, and SFP-A-Cur. Although the pore walls were dense in pure SF, adding pluronic to the polymeric solution induced micro-sized particulate morphology (Figure 1a). This confirms the role of pluronic as a surfactant that covers particles’ surfaces and induces a globular structure. The particle size and distribution of each sample containing pluronic was different, and the trend was SFP-A > SFP > SFP-A-Cur (Figure 1b–d). Regarding particle sizes, SFP-A was made of particles with a wide size distribution compared to SFP and SFP-A-Cur (Figure 1b–d). 

As depicted in Figure 1e, the cross-section of the samples (SFP-A-Cur) shows a gradient in the pore size, reducing in size from the bottom of the sample jar to the surface. The main reason of this hierarchical microstructure is the different levels of salt and solution in the jar. The solution in the bottom of the jar dried between the salt crystals, which induced larger pores. However, the solution on top of the jar, in the absence of salt crystals, underwent a simple casting procedure, which induces lower porosity with a smaller pore size. 

The FTIR characterization of different groups are displayed in Figure 2a. In the case of SF, three amide peaks of the β-sheet structure were detectable. Amide I was characterized by the C=O stretch at 1665 cm^−1^, amide II by the N-H band at 1540, and amide III by the C=N stretch at 1250 cm^−1^.

The –OH was also detectable as a wide peak in the range of 3000 cm^−1^–3650 cm^−1^ [37]. The main characteristic band in F127 was at the wavenumber of 1105 cm^−1^ which is related to C–O–C stretching vibrations. Curcumin also showed its fingerprint peaks at 3506 cm^−1^ (O-H stretching vibration), 1508 cm^−1^ (C=O and C=C vibrations), 1272 cm^−1^ (aromatic C–O stretching vibrations), and benzoate trans –CH vibration at 956 cm^−1^ [38,39]. Although most of the curcumin peaks were covered with the SF and F127 ones, almost every above-mentioned peak was detectable in the SFP-A-Cur sample. The 3506 cm^−1^ peak in curcumin is the result of free ortho position phenolic groups [40]. This peak disappeared in SFP-A-Cur, and a broad peak around 3290 cm^−1^ arose, which can be explained by the hydrogen bonding between curcumin, hydroxyl, and amide groups in the structure. The peak in 3290 cm^−1^ was previously observed in curcumin-Tween 80 which indicated the phenol-alcohol hydrogen bonds. Therefore, the adsorption of curcumin to the alcoholic groups in pluronic is more likely than the amide groups in SF [40].

To explain how curcumin absorption in the structure can affect the crystalline structure, the non-reversing heat flow spectrum from TMDSC for SFP, SFP-A, and SFP-A-Cur samples was plotted in Figure 2b. An exothermic peak in the range of 200–230 °C was also previously reported, which represents the random coil to β-sheet conformational transition in SF [41,42]. Considering the kinetic nature of this process, the characteristic peak can be recognized in the non-reversing term of heat flow. Importantly, β-sheet crystallization from residual random coil molecules of SF in SFP, SFP-A, and SFP-A-Cur samples occurred at different temperatures with different enthalpies (peak area). The trend for transition temperature was SFP-A-Cur > SFP > SFP-A while the enthalpy of transition was changed oppositely to SFP-A > SFP > SFP-A-Cur. According to the Gibbs free energy equation (∆G = ∆H − T∆S), changes in Gibbs free energy depend on the enthalpy and entropy changes and the temperature at which the transition takes place [43]. As the entropy in crystallization decreases (∆S < 0) and it is an exothermic transition (∆H < 0), the low transition temperature and the high absolute value of the enthalpy indicate a highly spontaneous and favorable condition for the transition (ΔG is a large negative number). Therefore, the crystallization affinity in the samples has the trend of SFP-A > SFP > SFP-A-Cur. This confirms that curcumin in the structure of SFP-A-Cur acted as a β-sheet inducer and consequently reduced the random coil residues. This can be explained by the hydrogen bonding between phenolic groups of curcumin and amide groups in silk. Silk fibroin has two main stable structures known as silk I and II. In silk I, the molecules have a mostly random and α-helix conformation containing the least crystallinity [30]. The structure of silk II is mostly formed of β-sheet conformation, whereas β-sheets are made of packed coils interacting with hydrogen bonds [19]. This crystalline structure contributes to the high strength and modulus properties of the SF-based materials. 

The stress–strain curves extracted from static compression tests showed that the required stress for applying constant strain increased exponentially. None of the samples collapsed after applying the net force of 18,000 N. The compression modulus and ultimate elongation were determined based on the stress–strain curves (Figure 2c). The higher average compression modulus of SF can be explained by the swelling ratio of the samples determined after five days. This value was calculated 818 ± 83%, 2692 ± 11%, 2605 ± 65%, and 1143 ± 31% for SF, SFP, SFP-A, and SFP-A-Cur, respectively. Lower water content in the SF sample can explain the higher average compression modulus. Although no significant difference in compression modulus was detected, the ultimate elongation increased by adding pluronic, acetone, and curcumin to the structure (Figure 2d,e). Nevertheless, considering the higher crystallinity in SFP-A-Cur, a higher compression modulus and lower elongation are expected for a single particle (dense structure) [30]. To explain the opposite trend seen in the globular scaffolds, it is necessary to understand the dynamics of wet granular systems. Different parameters such as particle size, polydispersity, and compacting ability of the particles affect the mechanical properties of such systems [44]. The capillary bridge between the particles is an important parameter which affects the mechanical properties of the hydrogels. As the particle size decreases, the capillary force and the bridge-breaking distance between the particles increase [44]. This physical fact can explain how SFP-A-Cur overcame high strains without breaking down. The other parameters affecting mechanical properties in compression are particle size polydispersity and single-particle stiffness, which have a direct impact on the compacting ability of the 3D structure [45]. In a system with size polydispersity, the coordination number of the particles and consequently the compacting ability increase. Soft particles with lower stiffness can also become more compact due to the transition ability to 3D polygons [45]. In the case of SFP-A-Cur, we expected higher stiffness on the single particles. In addition, considering the narrow size distribution in this sample, the packing ability of SFP-A-Cur is less than that of SFP or SFP-A. Competition of capillary forces and packing ability of the particles would accordingly determine the fate of the particles and the mechanical behavior of the 3D structure. As Figure 2e shows, SFP-A-Cur particles have tended to slip to the edges due to low packing ability, resulting in high strains. In comparison, the compression modulus showed more dependence on packing ability than capillary forces, which resulted in similar values to those of the globular samples. 

### 3.2. Degradation and Drug Release

The degraded residues were studied using optical microscopy (OM), and it confirmed that degradation of particulate 3D scaffolds of SFP, SFP-A, and SFP-A-Cur was mostly the result of microparticles separation (Figure 3a). 

The quantified weight loss (degradation %) showed the trend of SFP-A > SFP > SFP-A-Cur > SF for the samples for 14 days (Figure 3b). The degradation trend of SFP, SFP-A, and SFP-A-Cur was relative to the particle size, as the smaller particles provide higher surface area and interparticle interaction. In the wet granular systems, it also ends in a stronger capillary bridge and lower degradation rate [44]. This is in line with the trend of crystallization; the single particles’ stability enhances as the crystallinity increases [46]. The main degradation of all the samples occurred in the first three days and reached 60% for the curcumin-loaded hydrogel after 14 days (Figure 3b). The optimum biodegradation rate of the membranes used for tissue regeneration is correlated with wound closure rate. This value was previously reported around 60% in the presence of SF-based biomaterials [47]. For no-diabetic rats, the closure rate was also reported between 40 and 60% after 11 days [48]. Considering 55% degradation of SFP-A-Cur after 11 days, a suitable wound dressing performance was expected. The pH of the degradation media was also recorded for 14 days to determine the degradation mechanism (Figure 3c). As pluronic and its residues are neutral, the drop of pH proved the degradation of fibroin molecules [46]. 

The released curcumin from SFP-A-Cur was collected for 14 days. Although one phase association was the best-fitted kinetic model to explain the release profile (Figure 3c), the Korsmeyer–Peppas equation also provided a coefficient of determination (R-squared) of higher than 0.9. In the above-mentioned equations (Figure 3d,e), Q is the fraction of drug released at time t, k is the release rate constant, and n is the release exponent. The value of n in the Korsmeyer–Peppas model usually determines the mechanism of the release where n < 0.45 indicates that the drug diffuses partially through the matrix (microparticles) and water-filled pores. Once n equals to 0.45, a diffusion-controlled release from the microparticles is dominant, and values between 0.45 and 1.0 represent non-Fickian mass transport [49]. 

### 3.3. Cytotoxicity 

The cell viability of seeded cells on the surface of different experimental groups (Table 1) through MTT assay is shown in Figure 4a. The recorded OD after 1, 2, and 3 days showed no significant difference between groups. To see how the ratio of SF and pluronic may affect the cell viability, groups with different ratios of SF/P between 2 and 6 were also studied using MTT assay (Figure 4b). Non-significant difference between these groups was also confirmed. Figure 4c shows the morphology of the fibroblast cells with cytoskeleton stained after one and seven days of cell culture. The population of the cells increased after seven days in respect to day one. No significant difference in the morphology of the cells was observed which confirms no cytotoxicity among different groups of samples. Although SF lacks Arg-Gly-Asp (RGD), the main motif responsible for cell adhesion, it has shown acceptable cell adhesion [50,51]. Comparing the population of attached cells (day 1) to different experimental groups showed slight reduction after adding pluronic into the samples. This effect is caused by nonionic/nonpolar nature of pluronic and has been previously reported [52]. Despite this effect, the samples containing pluronic showed potential in promoting cell proliferation and consequently tissue regeneration.

### 3.4. Antibacterial Study

The CFU results for planktonic and adherent bacteria were plotted in Figure 5a,b. To evaluate the planktonic bacteria, all the data were normalized based on negative bacteria control, and for the adherent bacteria, the data were normalized based on the number of colonies attached to the pure SF sample. As Figure 5a shows, the number of planktonic bacteria increased in the presence of SF, while this number decreased in the presence of three other samples containing pluronic. This observation follows the reports on bactericidal and antifouling properties of pluronic 127. Studies showed this polymer inhibited planktonic bacteria by disrupting the lipid membrane of the cell [53]. As a surfactant, pluronic is able to prevent adhesion of the bacteria and consequently prevent biofilm formation [10,54]. Adding pluronic decreased planktonic bacteria up to 99.981% (4 logarithmic units) while curcumin could increase the antibacterial efficacy up to 99.999% (5 logarithmic units). The same trend was detected for the adherent bacteria. Comparing SFP and SFP-A samples with the control (SF) yields a 99.920% (3 logarithmic unit) decrease in the CFU count due to the presence of pluronic. In SFP-A-Cur, curcumin further increased the viable attached bacteria eradication up to 99.992% (4.5 logarithmic unit, Figure 5b) which is higher than similar studies [55,56]. There are different studies conducted on Gram-negative and positive bacteria to clarify the action mechanism of curcumin [57]. Various mechanisms were reported including inhibition of the bacterial Quorum sensing (QS) system, downregulation of the bacterial gene expression, blocking cell division, and bacterial cell membrane disruption that ends in the prevention of bacterial attachment [58]. Overall, based on the definition of MBC, SFP-A-Cur is considered antibacterial by ≥3 logarithmic unit reduction in planktonic and adherent bacteria [59].

To determine the pharmacodynamics of the samples as bacteriostatic or bactericidal materials, an indirect OD study was performed. Figure 5c represents the data collected over 800 min. It shows that there is a growth profile of *S. aureus* in the presence of SF, while the OD has been reduced in the supernatant collected from other experimental groups. In the case of SFP and SFP-A, the OD first dropped and then increased. To explain this trend, it is worth mentioning that the supernatant release from the globular samples is a dynamic system. As the degradation of these hydrogels is mostly due to the collapse of the particulate structure, the microparticles are found in the supernatant. Degradation of these particles during the incubation and OD recording affects the viability and proliferation of bacteria. Since the degradation of SFP-A is higher than SFP, a higher release of pluronic is expected over longer time periods. Therefore, it can be speculated why the minimum OD for SFP and SFP-A happened after 300 and 620 min. The OD and consequently the bacteria population decreased with a small negative slope of 0.006 unit/h for SFP-A-Cur, consistently. 

Because the cultured strain of bacteria was genetically fluorescent, the cultured samples were studied using CLSM after three times of washing with PBS (Figure 5d). The CLSM pictures and OD measurement collectively validate the CFU measurements. After all, the developed hydrogel containing curcumin showed antibacterial properties and cytocompatibility at the same time which makes it a good candidate for wound dressing and repair. Changing the ratio of SF/pluronic is also an effective parameter which can affect cellular and antibacterial properties. Increasing pluronic content as the surfactant agent is expected to decrease particle size and consequently degradation rate of the hydrogel. Different in vitro models have been previously developed to predict in vivo condition. However, the difficulty to run multi-cellular in vitro models, particularly with immune cells, necessitate the use of in vivo models to evaluate antibacterial properties and performance of the developed hydrogels [60].

## 4. Conclusions

In this study, the SF-based hydrogels with curcumin and globular microstructures were developed. Pluronic F127 is an amphiphilic molecule which changed the structure to particulate and provided antibacterial effects. The fabrication method of salt leaching/freeze-drying encouraged a globular structure and resulted in a pore size gradient. The contribution of curcumin in the blend of SF and pluronic was studied and it was confirmed that curcumin with phenolic groups interacted with SF as well as pluronic molecules through hydrogen bonds. Regarding the mechanical properties of the hydrogels, SFP-A-Cur microparticles showed higher ultimate elongation compared to other experimental groups. Degradation of the samples also correlated with the particle size, which is expected to be controllable as a function of SF/pluronic ratio. In the interface with fibroblast cells, curcumin-loaded hydrogels promoted cell attachment and proliferation. The studies on planktonic and adherent *S. aureus* bacteria showed pluronic and curcumin provided antibacterial effects. Pluronic added 4 and 3 logarithmic units (>99%) of antibacterial efficiency for planktonic and adherent bacteria, respectively. Adding curcumin into the hydrogels could increase this value up to 5 units. Overall, the collected data showed that although there are unmet properties, such as tissue adhesion, the novel developed hydrogel with a hierarchical structure can be used for infection-free skin wound dressing. For further improvement, the concentration of encapsulated curcumin can be optimized.

## Figures and Tables

**Figure 1 biomimetics-09-00483-f001:**
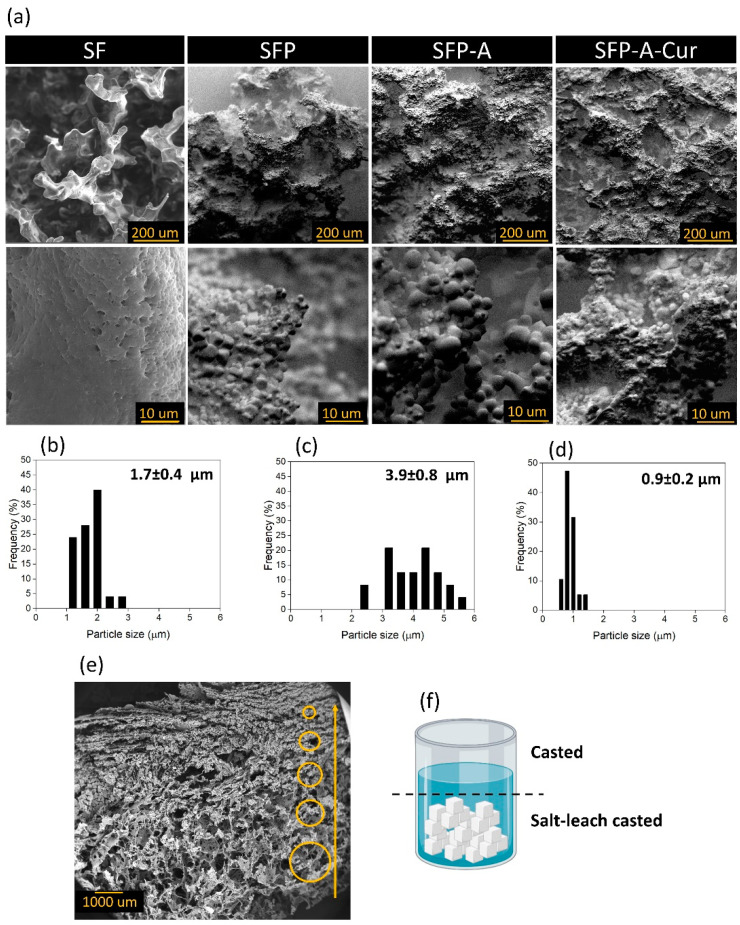
SEM micrographs of SF, SFP, SFP-A, and SFP-A-Cur in two magnifications (**a**), size distribution histograms of SFP (**b**), SFP-A (**c**), and SFP-A-Cur (**d**) extracted from the analysis of SEM images using ImageJ software. The cross-section of SFP-A-Cur with gradient pore size indicated with yellow circles in different sizes (**e**) which can be explained through the schematic of the solution after casting on the salt crystals (**f**). It shows how salt leaching is applied only in the bottom of the sample jar, while on the surface, a denser layer with a smaller pore size is cast (**f**).

**Figure 2 biomimetics-09-00483-f002:**
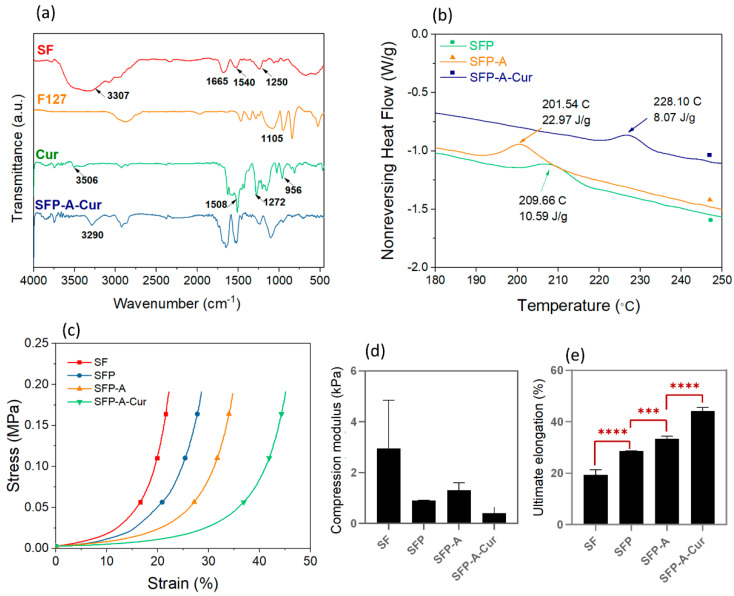
FTIR spectrum of SFP-A-Cur sample and all the compounds individually (**a**), and non-reversing heat flow vector extracted from TMDSC test of the samples containing pluronic, emphasizing on recrystallization temperature (**b**). The stress-strain plots of experimental groups were extracted from the uniaxial compression test (**c**) and extracted mechanical properties: compression modulus (**d**) and ultimate elongation *** *p* < 0.001, **** *p* < 0.0001 (**e**).

**Figure 3 biomimetics-09-00483-f003:**
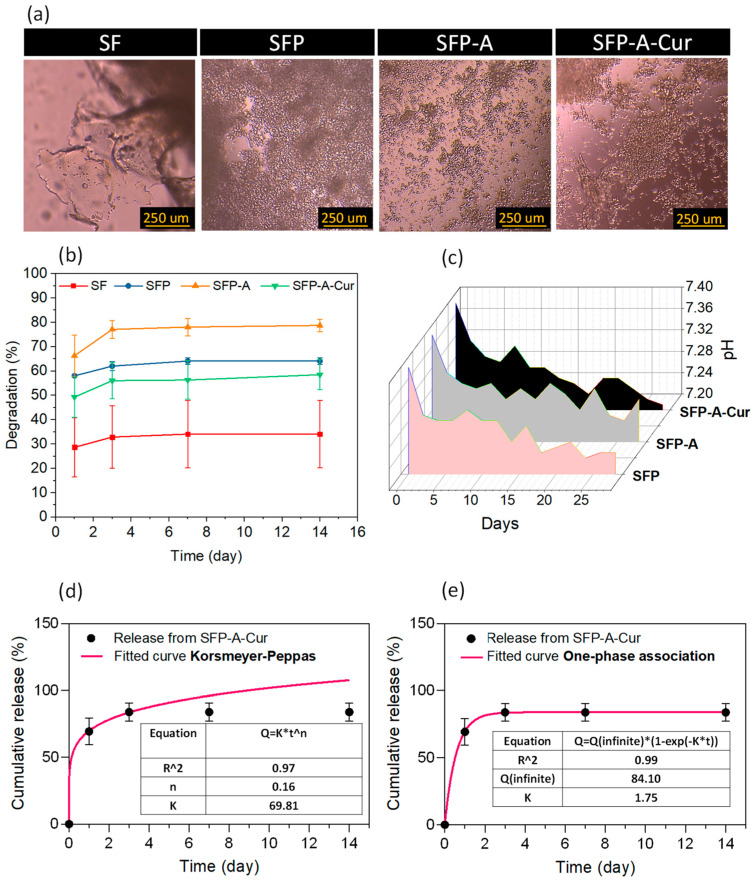
OM images of degredated residues in DMEM media (**a**), degradation profile based on weight loss (**b**) and pH variations (**c**) during 14 days of immersion in PBS for different experimental groups, and cumulative release of curcumin during 14 days calculated based on the absorption intensity of 400 nm wavelength, fitted with Korsmeyer–Peppas (**d**) and one-phase association (**e**) models.

**Figure 4 biomimetics-09-00483-f004:**
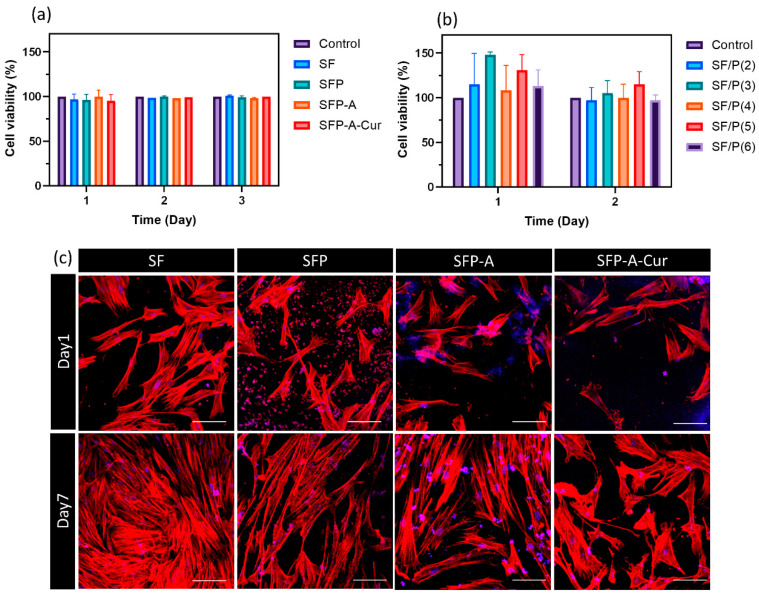
The viability of fibroblast cells cultured on the surface of four main experimental groups (**a**) and different groups with SF/P ratios between 2 and 6 (**b**). The reported values were normalized based on the control group at each time point—the CLSM images of cultured fibroblast cells stained with cytoskeleton assay. Actins are presented in red, and nuclei are colored in blue (the hydrogel is also colored in some parts with DAPI). Scale bar is equal to 100 µm (**c**).

**Figure 5 biomimetics-09-00483-f005:**
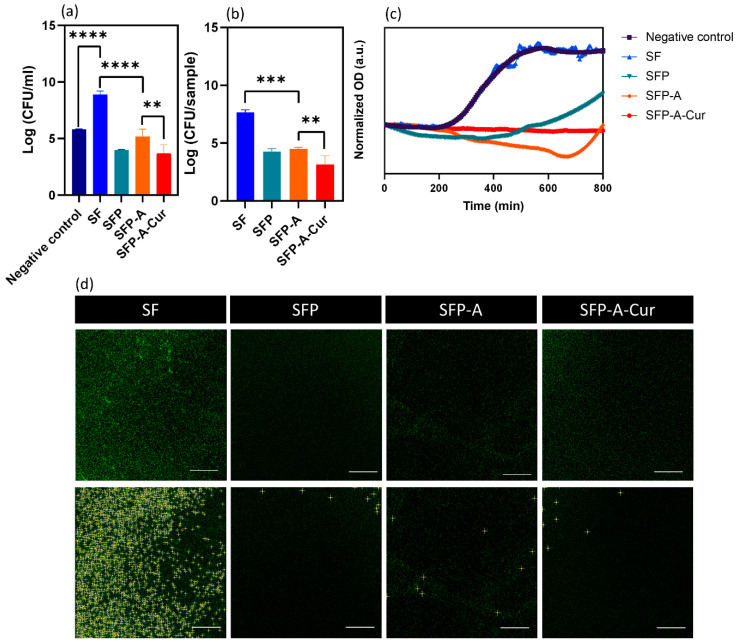
CFU results extracted from counted planktonic, ** *p* < 0.01, *** *p* < 0.001, **** *p* < 0.0001 (**a**) and adhesion (**b**) *S. aureus*, cultured on the surface of different experimental groups. The OD variations of bacteria suspension cultured in the curcumin release supernatant (**c**), showing the changes in bacterial growth. The CLSM images of the fluorescent *S. aureus* adhered to the surface (**d**), and the bright colonies are detected by ImageJ 1.54g based on prominence >8 in the lower row. Attached colony number extracted by ImageJ analysis.

**Table 1 biomimetics-09-00483-t001:** The solution composition to prepare the scaffolds for different groups.

Sample Code	Solution Composition (%*w*/*v*: g/mL)
SF	3% SF
SFP	3% SF–3% F127
SFP-A	3% SF–3% F127–5.3% acetone
SFP-A-Cur	3% SF–3% F127–5.3% acetone–0.04% curcumin

## Data Availability

Data are contained within the article. The raw data supporting the conclusions of this article will be made available by the authors on request.

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
