# Peer review of "Particulate 3D Hydrogels of Silk Fibroin-Pluronic to Deliver Curcumin for Infection-Free Wound Healing"

_biomimetics, 2024, doi:10.3390/biomimetics9080483_

Round 1

Reviewer 1 Report

Comments and Suggestions for Authors

In this manuscript, the author developed a spherical hydrogel composed of silk fibroin, pluronic F127, and curcumin. The effects of F127 and curcumin on the structure, mechanical properties, degradation rate, cell safety, and antibacterial ability of hydrogels were studied. However, there are some aspects that could be further clarified or addressed in the manuscript

1. What is the purpose of adding F127?

2. How to prove whether curcumin is uniformly dispersed in the hydrogel.

3. What is the adhesion and adhesion of hydrogel to skin in practical application?

4. There is an error in Abscissa in Figure 3d.

5. The antibacterial study section needs to refer to 10.1016/j.ijbiomac.2023.128496, and 10.1016/j.carbpol.2018.11.044 to enhance readability, especially for the exploration of anti-bacterial adhesion.

Author Response

Reviewer 1:

In this manuscript, the author developed a spherical hydrogel composed of silk fibroin, pluronic F127, and curcumin. The effects of F127 and curcumin on the structure, mechanical properties, degradation rate, cell safety, and antibacterial ability of hydrogels were studied. However, there are some aspects that could be further clarified or addressed in the manuscript

Comment 1: What is the purpose of adding F127?

Response 1: Thank you for pointing this out. We have previously mentioned the role of surfactants in theoretically homogenizing the distribution of hydrophobic molecules such as curcumin in hydrogels (line 70). On the other hand, F127 as a surfactant transformed the microstructure into a globular one (figure 1a) which later provided higher degradation rates in the range of wound closure rate (line 320-324). We added an extra reference which reports wound closure rate in SF-based biomaterials.

Comment 2: How to prove whether curcumin is uniformly dispersed in the hydrogel.

Response 2: While synthesizing the samples, it was visually apparent that increasing the amount of F127 improved the dispersion of curcumin in the final solution after adding curcumin solution. It is challenging, however, to prove how homogeneous its distribution is at the molecular level. One method is to compare the signal intensities in different sections of the hydrogel using fluorophore conjugated curcumin. Due to tight timing and complications with precise sectioning of the samples, we were unfortunately unable to perform this test.

Comment 3: What is the adhesion and adhesion of hydrogel to skin in practical application?

Response 3: This is a very interesting and important property that you have pointed out. Most studies in the literature evaluate the adhesion of different cell types to biomaterials. In spite of the absence of Arg-Gly-Asp (RGD), the common peptide motif responsible for cell adhesion, SF has demonstrated acceptable cell adhesion (Gholipourmalekabadi, M. et al. Silk fibroin for skin injury repair: Where do things stand? Adv Drug Deliv Rev 153, 28–53 (2020)). In addition to chemistry, cell adhesion also depends on the microstructure properties such as porosity and pore size. The CSLM imaging of fibroblast cells showed attachment of cells on SF sample, however addition of pluronic slightly decreased the population of the attached cells which is due to the non-polar non-ionic nature of pluronic (Vashi, A. V. et al. Adipose differentiation of bone marrow-derived mesenchymal stem cells using Pluronic F-127 hydrogel in vitro. Biomaterials 29, 573–579 (2008)) Over 7 days of study, cells attached to the pluronic containing samples proliferated, suggesting the potential of these samples to promote tissue regeneration. We added this discussion to section 3-3 (lines 350-355).

Adhesion of a biomaterial into the tissue happens through four mechanisms: mechanical interlocking, chemical bonding, diffusion theory and electrostatic theory. Therefore, SF lacks the necessary properties to interact with tissue. Multiple attempts have been made at modification of SF to increase its adhesion into the tissues such as skin as well as blending with PEG and  catechol group containing chemicals (Zheng, H. & Zuo, B. Functional silk fibroin hydrogels: preparation, properties and applications. J Mater Chem B 9, 1238–1258 (2021)). Therefore, we mentioned this gap in the conclusion (line 430).

Comment 4: There is an error in Abscissa in Figure 3d.

Response 4: Thank you, it is corrected.

Comment 5: The antibacterial study section needs to refer to 10.1016/j.ijbiomac.2023.128496, and 10.1016/j.carbpol.2018.11.044 to enhance readability, especially for the exploration of anti-bacterial adhesion.

Response 5: The requested references are added to the revised version. Please check reference 57 and 58 (line 380).

Reviewer 2 Report

Comments and Suggestions for Authors

·         It is not clear from the manuscript what is the resulted products – hydrogel film, lyophilized hydrogel particles or something else? For the degradation tests the samples were cut, whereas for antibacterial they were suspended. In Fig. 1 b, c, d the particle size distribution histograms are shown. However, for hydrogel films pore size is more important and should be estimated and compared. Addition of photo of the samples studied will improve understanding of the study for readers.

·         Introduction section should contain a clear explanation about biomimetic approach used in this study.

·         Materials and methods part. The description of the synthesis of the scaffolds is not provided sufficiently and does not allow to repeat the experiments. Please, check also the units.

·         The statement at line 320 that “The optimum biodegradation rate of the membranes used for tissue regeneration is correlated with wound closure rate” should be proved.

·         Fig. 3 d, e shows two fitting curves for the release profile. The curve in Fig. 3 e fits very well, whereas in Fig. 3 d does not. Why is the latter curve shown? Also standard deviation should be added. 

·         In the Abstract the authors state that “ pluronic by itself 30 caused more than 99% of planktonic and adherent antibacterial properties in the curcumin-free hydrogel groups.” The sentence is not clear. In the section Antibacterial study there are no results for solely pluronic.

Comments on the Quality of English Language

Extensive editing of English language required.

Author Response

Reviewer2:

It is not clear from the manuscript what is the resulted products – hydrogel film, lyophilized hydrogel particles or something else? For the degradation tests the samples were cut, whereas for antibacterial they were suspended. In Fig. 1 b, c, d the particle size distribution histograms are shown. However, for hydrogel films pore size is more important and should be estimated and compared. Addition of photo of the samples studied will improve understanding of the study for readers.

Comment 1: Introduction section should contain a clear explanation about biomimetic approach used in this study.

Response 1: Thank you for pointing this out. It was discussed how a pore size gradient can mimic full depth skin (lines 73 to 77).

“ As skin is a multilayered tissue, a hierarchical porous scaffold mimics the native tissue very well. In fact, the suitable pore size for neovascularization, fibroblast ingrowth and dermal repair was reported at 5, 5–15, and 20–125 μm, respectively [22] which shows the necessity of hierarchical porous scaffold use for skin tissue engineering.”

Comment 2: Materials and methods part. The description of the synthesis of the scaffolds is not provided sufficiently and does not allow to repeat the experiments. Please, check also the units.

Response 2: We added extra explanations regarding the ratios mentioned in the paper to clarify the synthesis process. More details about the process were also added to make it repeatable. Please check section 22 (lines 105 to 116).

Comment 3: The statement at line 320 that “The optimum biodegradation rate of the membranes used for tissue regeneration is correlated with wound closure rate” should be proved.

Response 3: This article focuses mainly on the design of the biomaterial and evaluation of its performance in simplified in vitro models, so in vivo research is beyond our current scope and must be carried out in a subsequent study. Nevertheless, we looked into the literature to find the numbers reported in different animal models that compare the degradation rate of our hydrogel with wound closure rate. For this purpose, references 48 and 49 (4 and 5 in this file) have been mentioned. “The optimum biodegradation rate of the membranes used for tissue regeneration is correlated with wound closure rate. This value was previously reported to be around 60% in the presence of SF-based biomaterials (Norahan, M. H., Pedroza-González, S. C., Sánchez-Salazar, M. G., Álvarez, M. M. & Trujillo de Santiago, G. Structural and biological engineering of 3D hydrogels for wound healing. Bioact Mater 24, 197–235 (2023)). A closure rate of 40-60% was also reported for no-diabetic rats after 11 days (Xu, Q. et al. Injectable hyperbranched poly(β-amino ester) hydrogels with on-demand degradation profiles to match wound healing processes. Chemical Science 9, 2179–2187 (2018)).”(line 328 to 329)

Comment 4: Fig. 3 d, e shows two fitting curves for the release profile. The curve in Fig. 3 e fits very well, whereas in Fig. 3 d does not. Why is the latter curve shown? Also standard deviation should be added.

Response 4: Thank you for your comment. A one-phase association fits the release data best, but it does not fit any other known release equations discussed based on different mechanisms such as diffusion, erosion and etc. Therefore, we report both the best fit from the known model (Korsmeyer-peppas) and the best fit from mathematical models (one phase association). The standard deviation bars in figure 3d and e have been added as you mentioned.

Comment 5: In the Abstract the authors state that “ pluronic by itself 30 caused more than 99% of planktonic and adherent antibacterial properties in the curcumin-free hydrogel groups.” The sentence is not clear. In the section Antibacterial study there are no results for solely pluronic.

Response 5: Thank you for the comment. Here we meant SFP sample (silk fibroin+ pluronic). To clarify the statement in the abstract we re-wrote the sentence “Interestingly, presence of pluronic caused more than 99% reduction in planktonic and adherent bacteria in the curcumin-free hydrogel groups” (line 32). You can find the related discussion on antibacterial properties on lines 369 to 377.

“As Figure 5a shows, the number of planktonic bacteria increased in the presence of SF, while this number decreased in the presence of three other samples containing pluron-ic. This observation follows the reports on bactericidal and antifouling properties of pluronic 127. Studies showed this polymer inhibited planktonic bacteria by disrupting the lipid membrane of the cell [54]. As a surfactant, pluronic is able to prevent adhe-sion of the bacteria and consequently prevent biofilm formation [55,56]. Adding pluronic decreased planktonic bacteria up to 99.981% (4 logarithmic units) while curcumin could increase the antibacterial efficacy up to 99.999% (5 logarithmic units). The same trend was detected for the adherent bacteria.”

Reviewer 3 Report

Comments and Suggestions for Authors

The authors propose a “Particulate 3D Hydrogels of Silk fibroin- pluronic to deliver 2 Curcumin for infection free wound healing”. Overall, the work is interesting, and the methodological part is quite complete. However, the results were not sufficiently presented and discussed in the study. Several doubtful points should be clarified.  Thus, I suggest that the paper should be subjected to a full revision.

Hereafter you can find general comments and detailed suggestions regarding the submitted manuscript:

1) SECTION 2.2: Provide additional information to clarify the gel production process. More in detail, how SFP-A was prepared? Is SF stable in acetone? Also, in the table 1 explaining the scaffold composition, formulation SF P2-6 were not present.

2) LINE 143: Add information about the curcumin calibration in PBS (did the authors add some excipients (also during in vitro release experiments) such as citric acid and ascorbic acid to stabilize Curcumin? Is Curcumin soluble in PBS? In which concentration range? Please also specify the range of linearity in the ABS).

3) ROW 153: Specify the cells used in the study

4) FIGURE 2: Correct Caption to figure 2 (f is missing). As for calorimetric results, no glass transition appeared in the thermograms presented so that it is difficult to understand the discussion of calorimetric study. Please also specify if the endothermic peaks are upward or downward.

5) ROW 250: the authors refer to image 2c, but it concerns the stress vs strain graph

6) ROW 251: Image 2b is for crystallization and not glass transition temperatures, so further considerations on thermodynamic analysis are necessary following the addition of the missing plot.

7) FIGURE 2(d): DS for SF too high

8) FIGURE 3(b): SD for SF too high

9) FIGURE 3 (d and e): Add the DS

10) LINE 320: Add bibliography

11) ROW 325: Is the pH drop (0.2) necessarily correlated to silk degradation? Please argue.

12) LINE 329: Add space "abovementioned"

13) SECTION 3.3: How was MTT test carried out for the samples in Table 1? (only SF/P(2)-SF/P(6) sample are mentioned in the methods)

14) How did the authors demonstrate that the increase in pluronic content led to a size decrease of the microparticles? Results are not convincing and/or are not well presented/discussed. Also, in my opinion the term microparticle is not adequate. The samples here presented appears as a porous scaffolds and the formation of separate microparticles is not demonstrated.

Comments on the Quality of English Language

English is fine, minor editing is required

Author Response

*please check the attached file, it includes a graph

Reviewer3:

The authors propose a “Particulate 3D Hydrogels of Silk fibroin- pluronic to deliver 2 Curcumin for infection free wound healing”. Overall, the work is interesting, and the methodological part is quite complete. However, the results were not sufficiently presented and discussed in the study. Several doubtful points should be clarified.  Thus, I suggest that the paper should be subjected to a full revision.

Hereafter you can find general comments and detailed suggestions regarding the submitted manuscript:

Comment 1: SECTION 2.2: Provide additional information to clarify the gel production process. More in detail, how SFP-A was prepared? Is SF stable in acetone? Also, in the table 1 explaining the scaffold composition, formulation SF P2-6 were not present.

Response 1: Thank you for your comment. We added extra explanations regarding the ratios mentioned in the paper to clarify the synthesis process. More details about the process were also added to make it repeatable. Please check section 2.2 (lines 108 to 118). We also discussed the preparation of SFP (2) to SFP(6). (lines 120 to 123).

Comment 2: LINE 143: Add information about the curcumin calibration in PBS (did the authors add some excipients (also during in vitro release experiments) such as citric acid and ascorbic acid to stabilize Curcumin? Is Curcumin soluble in PBS? In which concentration range? Please also specify the range of linearity in the ABS).

Response 2: No excipients has been used in release study and in vitro experiments. The calibration curve for curcumin release in PBS, pH=7.4 was determined between 0-75 ug/ml. The following figure presents linearity of absorption intensity at 400 nm.

Comment 3: ROW 153: Specify the cells used in the study

Response 3: Dermal fibroblasts were used for this purpose (line 157).

Comment 4: FIGURE 2: Correct Caption to figure 2 (f is missing). As for calorimetric results, no glass transition appeared in the thermograms presented so that it is difficult to understand the discussion of calorimetric study. Please also specify if the endothermic peaks are upward or downward.

Response 4: Thank you for your comment. We initially included the glass transition step in the manuscript, but later decided to exclude it. We corrected the caption and deleted the discussion about glass transition temperature from the revised version. Please find the discussion on lines 259 to 274.

“An exothermic peak in the range of 200-230 °C was also previously reported, which represents the random coil to β-sheet conformational transition in SF [42,43]. Consider-ing the kinetic nature of this process, the characteristic peak can be recognized in the non-reversing term of heat flow. Importantly, β-sheet crystallization from residual random coil molecules of SF in SFP, SFP-A and SFP-A-Cur samples occurred at differ-ent temperatures with different enthalpies (peak area). The trend for transition tem-perature was SFP-A-Cur>SFP>SFP-A while the enthalpy of transition was changed op-positely to SFP-A>SFP> SFP-A-Cur. According to the Gibbs free energy equation (∆G=∆H-T∆S), changes in Gibbs Free Energy depend on the enthalpy and entropy changes and the temperature at which the transition takes place [44]. As the entropy in crystallization decreases (∆S<0) and it is an exothermic transition (∆H<0), the low tran-sition temperature and the high absolute value of the enthalpy indicate highly sponta-neous and favorable condition for the transition (ΔG is a large negative number). Therefore, the crystallization affinity in the samples has the trend of SFP-A>SFP> SFP-A-Cur. This confirms that curcumin in the structure of SFP-A-Cur acted as a β-sheet inducer and consequently reduced the random coil residues.”

Comment 5: ROW 250: the authors refer to image 2c, but it concerns the stress vs strain graph

Response 5: The comment has been applied to the revised version.

Comment 6: ROW 251: Image 2b is for crystallization and not glass transition temperatures, so further considerations on thermodynamic analysis are necessary following the addition of the missing plot.

Response 6: The comment has been applied to the revised version.

Comment 7: FIGURE 2(d): DS for SF too high

Comment 8: FIGURE 3(b): SD for SF too high

Response 7, and 8: We think the pores size in pure SF is much bigger and randomly distributed than the samples containing pluronic that leads into more standard deviations.

Comment 9: FIGURE 3 (d and e): Add the DS

Response 9: The comment has been applied to the revised version.

Comment 10: LINE 320: Add bibliography

Response 10: Two references were added to the revised manuscript on line 328 (refs 48 and 49).

“This value was previously reported around 60% in the presence of SF based biomaterials [48]. For no-diabetic rats, the closure rate was also reported between 40-60% after 11 days [49].”

Comment 11: ROW 325: Is the pH drop (0.2) necessarily correlated to silk degradation? Please argue.

Response 11: According to the by-products of curcumin degradation (trans-6-(4′-hydroxy-3′-methoxyphenyl)-2,4-dioxo-5-hexenal, ferulic aldehyde, ferulic acid, feruloyl methane and vanillin) (Shen, L. & Ji, H.-F. The pharmacology of curcumin: is it the degradation products? Trends Mol Med 18, 138–144 (2012)), pH drops over time due to these 6 amino acids caused by fibroin degradation.

Comment 12: LINE 329: Add space "abovementioned"

Response: The comment has been applied to the revised version.

Comment 13: SECTION 3.3: How was MTT test carried out for the samples in Table 1? (only SF/P(2)-SF/P(6) sample are mentioned in the methods)

Response 13: The same methodology was applied for all the experimental groups (SF, SFP, SFP-A and SFP-A-Cur). It was mentioned in materials and methods (line 171-172).

Comment 14: How did the authors demonstrate that the increase in pluronic content led to a size decrease of the microparticles? Results are not convincing and/or are not well presented/discussed. Also, in my opinion the term microparticle is not adequate. The samples here presented appears as a porous scaffolds and the formation of separate microparticles is not demonstrated.

Response 14: As we have discussed in the first paragraph of section 3.1 (line 211-218), SEM images of the microstructures were analyzed using ImageJ software (mentioned in the caption of figure 1). However, no claim was made about pluronic content effect on microparticle size, however, based on their suggested formation mechanism, this effect is expected.  In fact, particle size was only compared between SFP, SFP-A and SFP-A-Cur samples.

The term “microparticle” was selected after collecting data on samples degradation. As Figure 3.1 shows, degradation of samples containing pluronic leaves microparticles as residues.

Reviewer 4 Report

Comments and Suggestions for Authors

Dear authors,

Thank you for your interesting article. Please refer to the following comments:

1.       For material specification please use the following term in brackets: (company, city, country)

2.       Determine CFU/ml instead of only CFU

3.       In the subheadings also define SF, SFP, SFP-A and SFP-A-Cur

4.       Define these abbreviations also in the M&M section and in the abstract

5.       Please discuss that for a measure to be called “antimicrobial” at least 3 log of bacterial reduction is necessary

6.       Refer to CENTER FOR DRUG EVALUATION AND RESEARCH, APPLICATION NUMBER: 208288Orig1s000

Author Response

Reviewer 4:

Thank you for your interesting article. Please refer to the following comments:

Comment 1: For material specification please use the following term in brackets: (company, city, country)

Response 1: Thank you for your comment. It was applied in section 2.1 of the revised version.

Comment 2: Determine CFU/ml instead of only CFU

Response 2: In this study, we have reported the antibacterial properties of the biomaterials against planktonic bacteria in CFU/ml. However, the antibacterial/antiadhesive properties against adherent bacteria were reported as CFU/sample to exclude the effect of PBS volume where samples were sonicated to detach the bacteria. In the discussion part, we used the word CFU refereeing to the methodology we used in this study.

Comment 3: In the subheadings also define SF, SFP, SFP-A and SFP-A-Cur

Comment 4: Define these abbreviations also in the M&M section and in the abstract

Response 3, and 4: The subheading and abstract do not contain any abbreviations. In the introduction, the abbreviation "SF" is first defined in line 79. Other sample codes are introduced in Table 1 within the Materials and Methods section, followed by their mention in the text. We do not understand the specific part you are referring to.

Comment 5: Please discuss that for a measure to be called “antimicrobial” at least 3 log of bacterial reduction is necessary

Response 5: The comment was applied to the revised version as follows:

“Overall, based on the definition of MBC, SFP-A-Cur is considered antibacterial by ≥3 logarithmic unit reduction in planktonic and adherent bacteria [61] “ (line 386-387).

Comment 6: Refer to CENTER FOR DRUG EVALUATION AND RESEARCH, APPLICATION NUMBER: 208288Orig1s000

Response 6: The document we found under this application number is related to “SoluPrep” which is an antiseptic solution for patient preoperative skin preparation. It is unclear which part of the manuscript is related and can be referred to this FDA regulatory approval.

Round 2

Reviewer 2 Report

Comments and Suggestions for Authors

1. The authors investigated influence of Pluronic F127 concentration in SF on the viability of fibroblast cells.  However, the cytotoxic and antibacterial properties of curcumin were not studied at different concentrations – only one curcumin concentration was used throughout the study – 0.04 %. The curcumin is a bioactive substance with strong antibacterial properties, so the SF samples with different concentrations of curcumin are worth to be studied.

2. What was the water content / moisture of the studied samples? Does the water content in SF samples affect the mechanical properties?

3. The authors use term “hydrogel” thought the article, whereas in Methodological (part 2.2. Synthesis of the scaffolds)  the authors mentioned that the obtained gels were lyophilized, and, presumably, in this form used for mechanical tests, cell viability study and antibacterial tests. The terminology used in the manuscript should be revised.

4. The details about sample form (hydrogel, lyophilized, powdered) used for the experiments should be provided in each methodological parts.

Comments on the Quality of English Language

Moderate editing of English language and style is required.

Reviewer 3 Report

Comments and Suggestions for Authors

REV 2

The authors propose a “Particulate 3D Hydrogels of Silk fibroin- pluronic to deliver 2 Curcumin for infection free wound healing”. Overall, the work is interesting, and the methodological part is quite complete. However, the results were not sufficiently presented and discussed in the study. Several doubtful points should be clarified.  Thus, I suggest that the paper should be subjected to a full revision.

Hereafter you can find general comments and detailed suggestions regarding the submitted manuscript:

Comment 1: SECTION 2.2: Provide additional information to clarify the gel production process. More in detail, how SFP-A was prepared? Is SF stable in acetone? Also, in the table 1 explaining the scaffold composition, formulation SF P2-6 were not present.

Response 1: Thank you for your comment. We added extra explanations regarding the ratios mentioned in the paper to clarify the synthesis process. More details about the process were also added to make it repeatable. Please check section 2.2 (lines 108 to 118). We also discussed the preparation of SFP (2) to SFP(6). (lines 120 to 123).

 ok

Comment 2: LINE 143: Add information about the curcumin calibration in PBS (did the authors add some excipients (also during in vitro release experiments) such as citric acid and ascorbic acid to stabilize Curcumin? Is Curcumin soluble in PBS? In which concentration range? Please also specify the range of linearity in the ABS).

Response 2: No excipients has been used in release study and in vitro experiments. The calibration curve for curcumin release in PBS, pH=7.4 was determined between 0-75 ug/ml. The following figure presents linearity of absorption intensity at 400 nm.

WHICH FIGURE?

Curcumin is poorly soluble in water, approximately 0.6µg/ml. According to the suppliers' technical specifications (Sigma-Aldrich), curcumin is soluble in a 0.5 M sodium hydroxide solution and subsequently diluted in PBS. Was this protocol used? If so, please specify.

Furthermore, without the use of antioxidant excipients, curcumin in solution is subject to rapid degradation. Considering the release time (14 days) the data may be unrealistic. Can the authors provide or have they monitored the UV spectrum of curcumin alone, used as a control, in the range of 200-800 nm in pbs 7.4. This could justify stability over time, up to 14 days.

Comment 3: ROW 153: Specify the cells used in the study

Response 3: Dermal fibroblasts were used for this purpose (line 157).

To which species do these cells belong (mouse, pig, human or other)? Please, specify the acronym  in material section.

Comment 4: FIGURE 2: Correct Caption to figure 2 (f is missing). As for calorimetric results, no glass transition appeared in the thermograms presented so that it is difficult to understand the discussion of calorimetric study. Please also specify if the endothermic peaks are upward or downward.

Response 4: Thank you for your comment. We initially included the glass transition step in the manuscript, but later decided to exclude it. We corrected the caption and deleted the discussion about glass transition temperature from the revised version. Please find the discussion on lines 259 to 274.

Why did the authors decide to exclude them? Aniway, in line 241 the reference to the Tg is still present.

“An exothermic peak in the range of 200-230 °C was also previously reported, which represents the random coil to β-sheet conformational transition in SF [42,43]. Consider-ing the kinetic nature of this process, the characteristic peak can be recognized in the non-reversing term of heat flow. Importantly, β-sheet crystallization from residual random coil molecules of SF in SFP, SFP-A and SFP-A-Cur samples occurred at differ-ent temperatures with different enthalpies (peak area). The trend for transition tem-perature was SFP-A-Cur>SFP>SFP-A while the enthalpy of transition was changed op-positely to SFP-A>SFP> SFP-A-Cur. According to the Gibbs free energy equation (∆G=∆H-T∆S), changes in Gibbs Free Energy depend on the enthalpy and entropy changes and the temperature at which the transition takes place [44]. As the entropy in crystallization decreases (∆S<0) and it is an exothermic transition (∆H<0), the low tran-sition temperature and the high absolute value of the enthalpy indicate highly sponta-neous and favorable condition for the transition (ΔG is a large negative number). Therefore, the crystallization affinity in the samples has the trend of SFP-A>SFP> SFP-A-Cur. This confirms that curcumin in the structure of SFP-A-Cur acted as a β-sheet inducer and consequently reduced the random coil residues.”

ok

Comment 5: ROW 250: the authors refer to image 2c, but it concerns the stress vs strain graph

Response 5: The comment has been applied to the revised version.

Looking at the slopes of the stress strain curve in figure c, it seems that there is an error in the values ​​of the compression modulus in figure d.

Comment 6: ROW 251: Image 2b is for crystallization and not glass transition temperatures, so further considerations on thermodynamic analysis are necessary following the addition of the missing plot.

Response 6: The comment has been applied to the revised version.

 ok

Comment 7: FIGURE 2(d): DS for SF too high

Comment 8: FIGURE 3(b): SD for SF too high

Response 7, and 8: We think the pores size in pure SF is much bigger and randomly distributed than the samples containing pluronic that leads into more standard deviations.

 ok

Comment 9: FIGURE 3 (d and e): Add the DS

Response 9: The comment has been applied to the revised version.

ok

Comment 10: LINE 320: Add bibliography

Response 10: Two references were added to the revised manuscript on line 328 (refs 48 and 49).

“This value was previously reported around 60% in the presence of SF based biomaterials [48]. For no-diabetic rats, the closure rate was also reported between 40-60% after 11 days [49].”

ok

Comment 11: ROW 325: Is the pH drop (0.2) necessarily correlated to silk degradation? Please argue.

Response 11: According to the by-products of curcumin degradation (trans-6-(4′-hydroxy-3′-methoxyphenyl)-2,4-dioxo-5-hexenal, ferulic aldehyde, ferulic acid, feruloyl methane and vanillin) (Shen, L. & Ji, H.-F. The pharmacology of curcumin: is it the degradation products? Trends Mol Med 18, 138–144 (2012)), pH drops over time due to these 6 amino acids caused by fibroin degradation.

A pH decrease of 0.2 seems a bit low to me.

Comment 12: LINE 329: Add space "abovementioned"

Response: The comment has been applied to the revised version.

ok

Comment 13: SECTION 3.3: How was MTT test carried out for the samples in Table 1? (only SF/P(2)-SF/P(6) sample are mentioned in the methods)

Response 13: The same methodology was applied for all the experimental groups (SF, SFP, SFP-A and SFP-A-Cur). It was mentioned in materials and methods (line 171-172).

In the cytotoxicity studies, neither the amount of platform tested nor the concentration of curcumin was reported. For antimicrobial studies the sample tested is 0.06 g: was it the same amount of sample also used for cytotoxicity studies? Please specify.

Comment 14: How did the authors demonstrate that the increase in pluronic content led to a size decrease of the microparticles? Results are not convincing and/or are not well presented/discussed. Also, in my opinion the term microparticle is not adequate. The samples here presented appears as a porous scaffolds and the formation of separate microparticles is not demonstrated.

Response 14: As we have discussed in the first paragraph of section 3.1 (line 211-218), SEM images of the microstructures were analyzed using ImageJ software (mentioned in the caption of figure 1). However, no claim was made about pluronic content effect on microparticle size, however, based on their suggested formation mechanism, this effect is expected.  In fact, particle size was only compared between SFP, SFP-A and SFP-A-Cur samples.

The term “microparticle” was selected after collecting data on samples degradation. As Figure 3.1 shows, degradation of samples containing pluronic leaves microparticles as residues.

In my opinion and observing SEM images, the sample appears as a crosslinked gel releasing microparticles after degradation. 
